# Kissing to Find a Match:
# Efficient Low-Rank Permutation Representation

**Hannah Dröge**
University of Siegen
57076 Siegen, Germany
hannah.droege@uni-siegen.de

**Zorah Lähner**
University of Siegen
57076 Siegen, Germany
zorah.laehner@uni-siegen.de

**Yuval Bahat**
Princeton University
Princeton, NJ 08544, United States
yuval.bahat@gmail.com

**Onofre Martorell**
University of Balearic Islands
Investigador ForInDoc del Govern de les Illes Balears
07122 Palma, Illes Balears, Spain
o.martorell@uib.cat

**Felix Heide**
Princeton University
Princeton, NJ 08544, United States
fheide@cs.princeton.edu

**Michael Möller**
University of Siegen
57076 Siegen, Germany
michael.moeller@uni-siegen.de

## Abstract

Permutation matrices play a key role in matching and assignment problems across the fields, especially in computer vision and robotics. However, memory for explicitly representing permutation matrices grows quadratically with the size of the problem, prohibiting large problem instances. In this work, we propose to tackle the curse of dimensionality of large permutation matrices by approximating them using low-rank matrix factorization, followed by a nonlinearity. To this end, we rely on the Kissing number theory to infer the minimal rank required for representing a permutation matrix of a given size, which is significantly smaller than the problem size. This leads to a drastic reduction in computation and memory costs, e.g., up to 3 orders of magnitude less memory for a problem of size $n = 20000$, represented using $8.4 \times 10^5$ elements in two small matrices instead of using a single huge matrix with $4 \times 10^8$ elements. The proposed representation allows for accurate representations of large permutation matrices, which in turn enables handling large problems that would have been infeasible otherwise. We demonstrate the applicability and merits of the proposed approach through a series of experiments on a range of problems that involve predicting permutation matrices, from linear and quadratic assignment to shape matching problems.

## 1   Introduction

Permutation matrices, which encode the reordering of elements, arise naturally in any problem that can be phrased as a bijection between two equally sized sets. As such, they are fundamental to many important computer vision applications, including matching semantically identical key points in images [50, 47, 48, 49], matching 3D shapes or point clouds [18, 46, 27], estimating scene flow on point clouds [35] and solving jigsaw puzzles [28], as well as to various sorting tasks [1, 16].

37th Conference on Neural Information Processing Systems (NeurIPS 2023).

$$2 \max \left( \begin{pmatrix} 1.0 & 0.0 \\ 0.5 & 0.87 \\ -0.5 & 0.87 \\ -1.0 & -0.0 \\ -0.5 & -0.87 \\ 0.5 & -0.87 \end{pmatrix} \cdot \begin{pmatrix} -1.0 & -0.0 \\ 1.0 & 0.0 \\ 0.5 & 0.87 \\ -0.5 & -0.87 \\ 0.5 & -0.87 \\ -0.5 & 0.87 \end{pmatrix}^{T} - \tfrac{1}{2}, 0 \right) = \begin{pmatrix} 0 & 1 & 0 & 0 & 0 & 0 \\ 0 & 0 & 1 & 0 & 0 & 0 \\ 0 & 0 & 0 & 0 & 0 & 1 \\ 1 & 0 & 0 & 0 & 0 & 0 \\ 0 & 0 & 0 & 1 & 0 & 0 \\ 0 & 0 & 0 & 0 & 1 & 0 \end{pmatrix}$$

Figure 1: Geometric intuition behind our approach on a 2D unit sphere. For well-distributed vectors $V \in \mathbb{R}^{\mathrm{Kiss}(2) \times 2}$, where the number of vectors is determined by the Kissing number ($\mathrm{Kiss}(2) = 6$), the cosine angle between different vectors $V_{i,:}$ and $V_{j,:}$, $i \neq j$, is $\langle V_{i,:}, V_{j,:} \rangle = \cos(\alpha) \leq 0.5$, while $\langle V_{i,:}, V_{i,:} \rangle = 1$ for the same vector. Thus, for any permutation $P$, the matrix-matrix product of $V$ and $(PV)^T$ merely has to be thresholded suitably to represent the permutation $P$, i.e. $P = 2 \max(V(PV)^T - 0.5, 0)$.

A permutation $p$, corresponding to the bijection from the set $\{1, \ldots, n\}$ onto itself, can be represented efficiently without a permutation matrix by merely enumerating the $n$ elements

$$(p(1), \ p(2), \ldots, p(n)) \in \mathbb{N}^n. \tag{1}$$

However, this representation is unsuitable for most computer vision problems that involve estimating $p$ through optimization since this representation (i) is inherently discrete, yielding combinatorial problems for which no natural relaxation exists, and (ii) induces a solution space with a meaningless distance metric, as element $i$ in the set generally is not 'closer' to element $i + 1$ than it is to any other element $j$.

As a result, almost all methods for predicting permutations, including learning-based methods, favor a *permutation matrix* representation instead, i.e., formulating a permutation as an element in the set

$$\mathcal{P}_n = \{ P \in \{0,1\}^{n \times n} \mid \sum_i P_{ij} = 1, \ \sum_j P_{ij} = 1 \ \forall i,j \}, \tag{2}$$

with $p(i) = j$ in representation (1) corresponding to $P_{ij} = 1$ in the matrix representation form (2), which allows predicting $p$ via optimization methods. Yet, the advantages of the matrix form representation (2) come at the cost of a prohibitive increase in memory, as it requires storing $n^2$ binary numbers $P_{ij} \in \{0,1\}$, or – after commonly used relaxations – even $n^2$-many floating point numbers instead of the $n$ integers in (1). This renders matching problems with $n > \ \sim 10^4$ largely infeasible as their corresponding permutation matrix $P$ constitutes over one hundred million entries.

To handle large matrices whose size prohibits explicit processing and storage, existing approaches typically either turn to *sparse representations*, i.e., storing only a small portion of matrix values in the form of $(i, j, P_{i,j})$ triplets, where $P_{i,j} \neq 0$, or employ *low rank representations*, i.e., forming a large matrix $P$ as a product of matrices

$$P = VW^T, \tag{3}$$

with $V, W \in \mathbb{R}^{n \times m}$ and $m << n$. Unfortunately, neither of these approaches is applicable to permutation matrices: sparse representation cannot be used as the sparsity pattern is not only unknown a-priori but actually the sought-after solution to the problem. On top of that, since permutation matrices are by definition full rank, a low-rank representation (3) can yield only a crude approximation at best.

**Contributions.** In this work, we alleviate the limitation on problem size by harnessing the well-studied problem of (bounds for) the so-called *Kissing number*, which, in practice, translates to introducing a simple adaptation to the matrix factorization approach (3). In particular, we exploit the fact that for row-normalized matrices $V$ and $W$, the entries of $VW^T$ correspond to the cosines of the angles between the matrix rows. We then apply a pointwise non-linearity on the product of the matrices in (3), which allows representing any permutation while using $m << n$. We use the Kissing number theory to provide an estimate for how small an $m$ we can use. We elaborate on these theoretical considerations in Section 3 and provide an illustration of the geometric intuition for our approach in Fig. 1.

We then demonstrate the applicability of the proposed approach through several numerical experiments tackling various problems that involve estimating permutations, including a study on point alignment, linear assignment problems (LAPs), quadratic assignment problems (QAPs), and a real-world shape matching application. We find that the proposed approach trades off only little accuracy to offer significant memory saving, thus enabling handling bijection mapping problems that are larger than was previously possible, when full permutation matrices had to be stored.

## 2   Related Work

### 2.1   Permutation Learning and Representation

Permutation learning aims to develop a model capable of predicting the optimal permutation matrix that matches a given input data or label. Previous studies have suggested relaxing permutation matrices to continuous domains, such as the Birkhoff polytope [2], to approximate solutions for these optimization problems but still faced the problem of enforcing the sum-to-one constraints, which are commonly approximated using Sinkhorn layers [34]. In 2011, Adams *et al.* [34] proposed the Sinkhorn propagation method, with the goal of iteratively learning doubly-stochastic matrices that allow the prediction of permutation matrices. Following the same research line, Cruz *et al.* [38] proposed Sinkhorn networks, followed by Mena *et al.* [28] proposing Gumble-Sinkhorn networks, adding Gumbel noise to the Sinkhorn operator to learn latent permutations. Grover *et al.* [16] made further efforts in the relaxation of a learned permutation matrix and propose to relax the matrix to become unimodal row-stochastic, meaning that the rows must maintain a total sum of one while having a distinct argmax. More recent studies suggest to circumvent the constraint of row and column sums being one by predicting permutations in Lehmer code, whose matrix form results to be row-stochastic [9]. Other works propose alternative representations of permutations in different domains, as to work in the Fourier domain to give a compact representations of distribution over permutations [17, 20], or embed them to the surface of a hypersphere [33].

### 2.2   Nonlinear Matrix Decomposition

In previous work on nonlinear matrix decomposition, Saul [40] aims to find a low-rank matrix $L$ that approximates a sparse nonnegative matrix $P$ by applying an elementwise nonlinearity $\sigma$ to $L$, i.e. $P \approx \sigma(L)$, and proposed an alternating minimization method to solve for $L$ directly. In further work, [39], Saul analyzed the geometric relationship between the sparse and low-rank matrices, specifically for $\sigma$ as a rectified linear unit (ReLU). For ReLU as nonlinearity, Seraghiti *et al.* [42] subsequently proposed an accelerated method of [40] to solve this problem, and a method, working with a version on $L = VW$, derived from the product of two matrices. In our work, we exploit this problem of low-rank approximation for permutation matrices and show its relationship to the Kissing number.

### 2.3   Assignment Problems

The goal of assignment problems is to find a permutation between two ordered sets while minimizing the assignment costs. The two most common versions are the *linear* and *quadratic* assignment problem (LAP and QAP) which are based on element-wise and pair-wise costs, respectively. An LAP can be solved in cubic time using the Hungarian algorithm [22]. The QAP was first introduced by Koopmans and Beckmann [21] and can be written as $\min_{P \in \mathcal{P}_n} \operatorname{tr}(APBP^\top) + \operatorname{tr}(C^\top P)$, for $A, B \in \mathbb{R}^{n \times n}$ and $C \in \mathbb{R}^{n \times n}$ being a problem where $A \in \mathbb{R}^{n \times n}$ is the cost function between elements of the first object to match, $B \in \mathbb{R}^{n \times n}$ is the distance function between elements in the second object and $C \in \mathbb{R}^{n \times n}$ is a point-wise energy. The QAP has been proved to be NP-hard so no polynomial time solution can be expected for general cases. As a result, many relaxations of the problem exist, for example by relaxing the permutation constraint [14, 36], or by lifting the problem to higher dimensions [18, 51]. A survey on various relaxation approaches can be found in [24]. While the relaxations do ease some aspects of the problems, they normally do not decrease the dimensionality of the problem which remains demanding for large $n$.

### 2.4   3D Shape Correspondence

3D shape correspondence is also often posed as an assignment problem between the sets of vertices of a pair of shapes, for example through point descriptors matched by an LAP or in an QAP aiming to

preserve distances between all point pairs. However, 3D shapes are often discretized with thousands of vertices which makes optimization for a permutation computationally challenging. Hence, the permutation constraint is often relaxed [36] and, even though the tightness of relaxation might be known [1, 10], the optimization variables still scale quadratically with the problem size. In [13] and [46] the QAP is deconstructed into smaller problems and then each of them is optimized with a series of LAPs, while [41] solve for permutations as a series of cycles that gradually improve the solution.

Because permutation constraints for large resolution become infeasible, and, hence, the restriction to cases with the same number of vertices, recent methods often do not impose these constraints at all. The functional maps framework [30] converts the output to a mapping between functions on the shapes instead of vertices and can reduce the dimensionality drastically by using the isometry invariance property of the eigenfunctions of the Laplace-Beltrami operator [32]. Other lines of work rely on a given template to constrain the solution [15, 44], impose physical models that regularizes the deformation between inputs [11, 12, 31], or learn a solution space through training [8, 23, 26]. However, with the exception of template-based models, these cannot guarantee permutations.

## 3   Low-Rank Permutation Matrix Representation

A common approach to solve optimization problems with costs $E$ over the set of permutation matrices $\mathcal{P}_n$ (including those arising from training neural networks for predicting assignments) is to relax the problem by replacing $\mathcal{P}_n$ by its convex hull conv($\mathcal{P}_n$), i.e., the set of doubly-stochastic matrices:

$$\min_{P \in \text{conv}(\mathcal{P}_n)} E(P). \tag{4}$$

Since $P$ grows quadratically in $n$, has an unknown sparsity pattern, and the true solution is always full rank, such problems pose significant challenges for large $n$. In this work, we make the interesting observation that a non-linearity as simple as a rectified linear unit (ReLU, denoted by $\sigma$) is sufficient not only to restore a full rank, but to represent any permutation matrix exactly. More precisely, we propose to replace the set conv($\mathcal{P}_n$) in (4) with the set $\mathcal{K}_m(\mathcal{P}_n) = \{\sigma(2VW^T - 1) \mid V, W \in \mathbb{R}^{n \times m}\}$ and use the so-called *Kissing number* [4, 29, 52] to show that $\mathcal{P}_n \subset \mathcal{K}_m(\mathcal{P}_n)$ for a surprisingly small $m$. Let us first formalize our approach by defining the *Kissing number*:

**Definition 1.** *For a given $m \in \mathbb{N}$, we define the Kissing number Kiss($m$) as*

$$Kiss(m) := \max_n \{n \in \mathbb{N} \mid \exists A \in \mathbb{R}^{n \times m}, \|A_{i,:}\|_2 = 1, 2\langle A_{i:}, A_{j,:} \rangle \leq 1, i \neq j\}. \tag{5}$$

Note that the Kissing number can be interpreted geometrically as the maximum number of points that can be distributed on an $m$-dimensional unit sphere such that the angle formed between each pair of different points is at least $\arccos(0.5)$. This property quickly establishes $\mathcal{P}_n \subset \mathcal{K}_m(\mathcal{P}_n)$ :

**Proposition 1.** *Let $P \in \mathcal{P}_n$ be an arbitrary permutation matrix, and let $\sigma$, $\sigma(x) = \max(x, 0)$ denote a rectified linear unit (ReLU). Then for every $m$ such that $n \leq Kiss(m)$ there exist $V, W \in \mathbb{R}^{n \times m}$ such that*

$$P = \sigma(2VW^T - 1). \tag{6}$$

*Proof.* Let $V \in \mathbb{R}^{n \times m}$ be a matrix that satisfies the equalities and inequalities of (5), and let $W = PV$. Then it holds that

$$2\langle V_{i,:}, W_{j,:} \rangle \begin{cases} \leq 1 & \text{if } P_{i,j} \neq 1 \\ = 2 & \text{otherwise} \end{cases}. \tag{7}$$

Consequently

$$\sigma(2\langle V_{i,:}, W_{j,:} \rangle - 1) = \begin{cases} 0 & \text{if } P_{i,j} \neq 1 \\ 1 & \text{otherwise} \end{cases}, \tag{8}$$

which proves the assertion. $\qquad\square$

To determine the minimal rank $m$ that is required for representing a permutation of $n$ elements, we rely on extensive studies in the past few decades which computed either exact values or lower and upper bounds for different values of $m$ [7].

Using $P = \sigma(2VW^T - 1)$ for relaxing (4) yields a relaxation that requires only $2mn$ instead of $n^2$ parameters, with $m << n = \text{Kiss}(m)$. For instance, $\text{Kiss}(24) = 196560$ implies that matrices of rank $m = 24$ are sufficient for representing any arbitrary permutation matrix of up to $n = 196560$ elements, thus requiring $\sim 4000$ times less storage memory: $2 \cdot 24 \cdot 196560$ instead of $196560^2$ parameters. Furthermore, $\mathcal{P}_n \subset \mathcal{K}_m(\mathcal{P}_n)$ ensures that – in stark contrast to direct low-rank factorization – *any* permutation matrix can still be represented *exactly*. Empirically, the optimization over parametrizations $\sigma(2VW^T - 1)$ turned out to cause significant challenges, likely due to the non-convexity and non-smoothness of the problem. To alleviate this problem, we resort to a smoother version of (6) which can still approximate permutations to an arbitrary desired accuracy:

**Proposition 2.** *Let $P \in \mathcal{P}_n$ and $g$ denote an arbitrary permutation matrix and an arbitrary entry-wise strictly monotonically increasing function, respectively, and let $s$ denote the row-wise Softmax function $s(A)_{i,j} = \frac{\exp A_{i,j}}{\sum_k \exp A_{i,k}}$. Then $\forall n \leq \text{Kiss}(m)$ and $\forall \epsilon > 0$ there exist $V, W \in \mathbb{R}^{n \times m}$ and $\alpha > 0$, such that*

$$\|P - s\left(\alpha g(VW^T)\right)\| \leq \epsilon.$$

*Proof.* Similar to the proof in Proposition 1, we start by choosing $V$ satisfying (5) and setting $W = PV$ to obtain

$$(VW^T)_{ij} = \langle V_{i,:}, W_{j,:} \rangle \begin{cases} = 1 & \text{if } P_{i,j} = 1 \\ \leq 0.5 & \text{otherwise} \end{cases}.$$

Then $\forall i, j, k$ s.t. $P_{ij} = 1$ and $k \neq j$ (i.e., $P_{ik} = 0$) it holds that $g(VW^T)_{ij} > g(VW^T)_{ik}$. Finally, to yield the assertion we use the Softmax property of converging to the unit vector in the limit $s(\alpha A_{i,:}) \overset{\alpha \to \infty}{\to} e_j$ (with $j = \arg\max A_{i,:}$), by taking $\alpha > 0$ to be large enough. $\square$

In practice, we use $g(x) = 2x$, in accordance with the representation in (6). We use this smoother version to validate the proposed low-rank representation for handling large matching problems in the experiments we report next. Please notice that this is only a proof of existence of a decomposition into $V, W$, not every $V, W$ lead to a permutation and an optimisation is not guaranteed to find a correct decomposition.

## 4 Experiments

The following experiments validate our efficient permutation estimation method for different applications, and they confirm the ability to scale to very large problem sizes. First, as a proof of concept, we demonstrate our approach on the application of point cloud alignment for the two non-linearities proposed in Section 3 and introduce our sparse training technique. We then validate the effectiveness of our approach in the context of linear assignment problems and show how to handle sparse cost matrices. We perform further experiments in the context of generic NP-hard quadratic assignment problems, and integrate our approach into a state-of-the-art shape matching pipeline, thus providing the same level of accuracy while enabling a higher spatial resolution.

### 4.1 Implementation Details

We use the PyTorch Adam optimizer [19] with its default hyperparameters in all our experiments. The estimation of $V$ and $W$ is performed in parallel.

**Stochastic Optimization.** Fully benefiting from our proposed compact representation requires the costs $E$ (or an approximation thereof) to be evaluated *without ever forming the full (approximate) permutation matrix*, as this step would inherently return to necessitate $n^2$ many entries. To this end, we introduce the concept of *stochastic optimization*, which – for our softmax-based representation $s(2\alpha VW^T)$ arising from Proposition 2 – is not a stochastic training in a classical sense: we propose to fix all but two entries in each row of our approximate permutation. Specifically, in any supervised (learning-based) scenario where it is known that the $y_i$-entry of the $i$-the row of the final permutation $P$ ought to be equal to one, each step of our optimizers merely computes the $y_i$-th and one randomly chosen ($r_i$-th entry) of each row, and computes the softmax $s$ on these two entries only while implicitly assuming $P_{i,j} = 0$ for $j \notin \{y_i, r_i\}$, i.e.,

$$P_{i,[y_i,r_i]} = s(2\alpha V_{i,:}(W_{[y_i,r_i],:})^T). \tag{9}$$

In the above, we used $W_{[y_i, r_i],:}$ to denote the $2 \times m$ matrix consisting of the $y_i$-th and the $r_i$-th row of $W$. Our stochastic approach requires the computation of $2n$ entries per gradient descent iteration only and – by randomly choosing the $r_i$ – manages to still approximate the desired objective well.

**Normalization of $V$ and $W$.** Since propositions 1 and 2 rely on row-normalized matrices, we explicitly enforce this constraint whenever we compute $P$, by using $V_{i,:} \leftarrow \frac{1}{\|V_{i,:}\|} V_{i,:}$, $W_{i,:} \leftarrow \frac{1}{\|W_{i,:}\|} W_{i,:}$. We omit this step from the presentations below for the sake of readability.

**Softmax Temperature.** Since the values of $\langle V_{i,:}, W_{j,:} \rangle$ are bounded by one following the aforementioned normalization, the *temperature* parameter $\alpha$ determines the trade-off between approximating a hard maximum (as required for accurately representing permutations, see Prop.2) and favorable optimization properties (i.e., meaningful gradients). We specify the schedule (constant or monotonically increasing) in each of the experiments below.

## 4.2 Point Cloud Alignment

As a proof of concept, we demonstrate that our proposition is correct and the optimization process converges. We explore the different choices of non-linearity, starting with ReLU and continuing with Softmax, using the task of predicting a linear transformation over point clouds. In this task we aim to match a point cloud $X_1 \in \mathbb{R}^{n \times m}$ consisting of $n$ $m$-dimensional points, uniformly distributed on the unit hyper-sphere, to its linearly transformed and randomly permuted version $X_2 \in \mathbb{R}^{n \times m}$. To obtain $X_2$ we multiply $X_1$ by a randomly drawn matrix $\Theta_{\text{GT}} \in \mathbb{R}^{m \times m}$ and apply a random permutation. Then, we optimize over the estimated transformation matrix $\Theta$ which in this experiment defines our permutation matrix $P(\Theta)$:

$$P(\Theta) = \sigma \left( 2VW(\Theta)^T - 1 \right). \tag{10}$$

Note that our representation in (6) is fully parameterized by $\Theta$, with $V = X_1$ and $W(\Theta) = X_2\Theta$, and $V$ and $W$ are row-wise normalized in each iteration. Here $P(\Theta)$ is equal to the correct permutation if $\Theta$ correctly aligns the point clouds, i.e. minimizes the angle between corresponding points in $V$ and $W(\Theta)$ while maximizing the angles between non-corresponding points.

We solve for the permutation by performing 20000 minimizing steps with a learning rate set to 0.01 over the negative log-likelihood loss

$$\hat{\Theta} = \arg\min_{\Theta} \ -\frac{1}{n} \sum_{i=1}^{n} \log\left( P(\Theta)_{i, y_i} \right), \tag{11}$$

where $y_i$ is the index of the point in $X_2$ which corresponds to the $i^{\text{th}}$ point in $X_1$. We experiment with different numbers of points $n$, each time choosing the dimension $m$ to be just big enough to satisfy the Kissing number constraint from Proposition 1, i.e., $\text{Kiss}(m) \geq n > \text{Kiss}(m - 1)$. To check that we were able to find the correct transformation matrix $\Theta$ – and therefore the correct permutation matrix $P$ – through optimization, we verify that the nearest neighbor (closest point) for each row $i$ in $V$ is located in row $j$ of matrix $W$ that satisfies $P_{i,j} = 1$. We find that this is indeed the case in all experiments with different numbers of points $n \in \{10, 100, 1000, 10000\}$, thus establishing that we could reach the correct permutation through optimization. We achieve equally good results when replacing the point-wise non-linearity ReLU with Softmax $P(\Theta) = s\left( 2\alpha VW(\Theta)^T \right)$.

Due to the quadratically growing size of the permutation matrix with an increasing number of points, we further propose to optimize for the permutation matrix stochastically, as described in Section 4.1. We ran experiments with similar settings as above, wherein we gradually increased the value of the temperature parameter $\alpha$ linearly during optimization from $\alpha = 5 \cdot 10^{-5}$ to $\alpha = 1000$. In these experiments, we again found that each point was paired with its corresponding nearest neighbor. Also, we could reduce the memory consumption, as shown in Fig. 2.

## 4.3 Point Cloud Alignment on Spectral Point Representation

We conduct an additional experiment on point cloud alignment in the context of the functional maps framework [30]. Here, the goal is to extract a point-to-point correspondence between two shapes $X, Y$ from a $m \times m$-dimensional functional map $C$ [30] where $m$ is much smaller than the number of vertices in $X$ and $Y$. A possible interpretation of $C$ is that it aligns the spectral point representations $\Phi_X, \Phi_Y \in \mathbb{R}^{n \times m}$ in which each point $x$ is represented by the vector of values

of the first $m$ Laplace-Beltrami eigenfunctions at $x$ such that $P \cdot \Phi_X \approx \Phi_Y \cdot C$ where $P$ is the unknown permutation between $X$ and $Y$. Given $C$, $P$ can be retrieved by a nearest-neighbor query between $\Phi_X, \Phi_Y C$, as proposed in the original paper (see [30], Section 6.1), or by solving a Linear Assignment Problem (LAP) if a bijection is desired. This is exactly the same setting as in in Section 4.2 with a small amount of noise in the point clouds. We use the FAUST registrations [3] with the original 6890 vertices, a downsampled version to 502 vertices for those experiments and $C$ generated by the ground-truth correspondence. Then, $\Phi_X, \Phi_Y C$ can be directly used for $V$ and $W$ in our method.

We compare our method, which calculates correspondences the same way as described in Section 4.2, to a general LAP solver (specifically the Jonker-Volgenant algorithm from sklearn.linear_sum_assignment), nearest neighbor computation, optimal transport (as implemented in the python POT package), and stochastic matrices generated by Sinkhorn iterations. In Table 1 we show that our method outperforms all baselines in terms of geodesic error of the final matching and shows positive trends in terms of runtime and memory consumption.

| | 502 vertices, $m = 20$ | | | 6890 vertices, $m = 50$ | | |
|---|---|---|---|---|---|---|
| Method | Error | Time | Memory | Error | Time | Memory |
| LAP | $1.3 \times 10^{-1}$ | 0.023s | 8.02 MB | $3.1 \times 10^{-1}$ | 79.2s | 565.31 MB |
| Nearest-Neighbors | $8.2 \times 10^{-1}$ | 0.008s | 5.22 MB | $4.0 \times 10^{-1}$ | 2.6s | 34.00 MB |
| Optimal Transport | $3.8 \times 10^{-1}$ | 0.524s | 12.58 MB | $2.0 \times 10^{-1}$ | 182.7s | 1862.27 MB |
| Sinkhorn iterations | $1.4 \times 10^{-1}$ | 0.030s | 9.4 MB | $3.0 \times 10^{-1}$ | 12.0s | 750.55 MB |
| **Ours** | $2.2 \times 10^{-3}$ | 0.801s | 18.18 MB | $2.5 \times 10^{-2}$ | 77.6s | 42.35 MB |

Table 1: Comparisons of point-wise correspondence extraction from ground-truth functional map [30] for FAUST. The error is the mean geodesic matching error of all points. Please note that all code except ours and Sinkhorn iterations are from libraries that are likely more optimized in terms of runtime and memory consumption. The memory consumption is evaluated on CPU only.

## 4.4 Linear Assignment Problems

We next validate our method on balanced Linear Assignment Problems (LAPs), which typically involve assigning a set of agents to an equally sized set of tasks, e.g., when optimizing the allocation of resources. We show results on a collection of regularized LAPs in the form

$$\arg\min_{V,W} \quad \underbrace{\mathbf{tr}(A \cdot P(V,W))}_{\text{LAP term}} + \underbrace{\mu(P(V,W))}_{\text{regularizer}}, \tag{12}$$

where $P(V,W) = s\left(2\alpha VW^T\right)$ is a permutation and $A \in \mathbb{R}^{n \times n}$ is some given similarity matrix. While the Softmax non-linearity ensures all rows sum to one, $\mu(P(V,W))$ is a regularization term enforcing columns summing to one as well, to satisfy the permutation constraints:

$$\mu(P) = \sum_j \left(\sum_i P_{ij} - 1\right)^2. \tag{13}$$

Due to the row-wise Softmax all rows already sum to one but we incentive the columns to sum to one as well, as is necessary for permutations.

**Dense Matrices.** We evaluate on a set of LAPs based on descriptor similarities of 3D shapes from the FAUST dataset [3], with $n$ randomly chosen vertices per object [13]. Let $D_X, D_Y \in \mathbb{R}^{n \times k}$ be two k-dimensional point-wise descriptors of the shapes $X, Y$ corresponding to $n$ points. We use the SHOT [37] ($\mathbb{R}^{n \times 352}$) and the heat kernel signature [43] ($\mathbb{R}^{n \times 101}$) with their default parameters as descriptors and stack them together to comprise $D. \in \mathbb{R}^{n \times 453}$ in total, then $A = D_X \cdot D_Y^\top$. Solving an LAP with this type of similarity matrix $A$ is used e.g. in [46] as the initialization strategy.

We generate 100 problem instances by pairing each of the 100 shapes in FAUST with a random second shape to get the pair $X, Y$. Then we evaluate the relative error of the energy compared to the energy of the optimal solution (restricted to valid permutation solutions), and the average Hamming distance to the next valid permutation (namely, the number of elements that violate the permutation constraint). We ran experiments with $n = 100, m = 30, \alpha = 20$ and used a greedy heuristic to generate valid

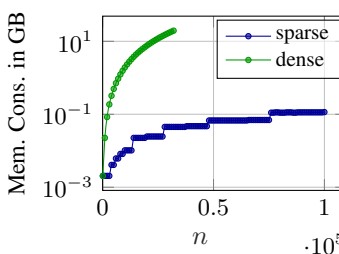
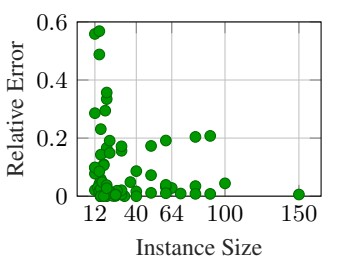
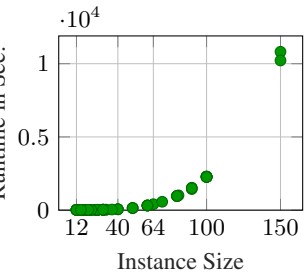

Figure 2: Memory consumption for point cloud alignment

Figure 3: Relative error and runtime on QAPLIB dataset

permutations from the results violating the permutation constraint (iteratively projecting the maximum value of the permutation to one, and the rest of the corresponding row and column to zero). Out of the 100 instances, 53 lead to valid permutations without the heuristic. The average relative error of immediately valid permutation is $1.8\%$ and after pseudo-projection of all instances it is $2.0\%$. Due to the Softmax, every matrix has 100 non-zero entries that are all nearly equal to one. On average, the Hamming distance of invalid permutations to the next valid one is $1.38$ ($1.4\%$ of the problem size) which means in most cases one would have a valid permutation after adjusting one entry.

**Sparse Matrices.** Given a matrix $A$, that is sparsely populated and only contains non-zero entries in a subset $S = \{(i,j)|A_{i,j} \neq 0\}$, we compute and optimize the permutation matrix sparsely in $(i,j) \in S$ by calculating the matrix factorization only at the required entries, similar to Section 4.1, but without restricting the number of entries per row of $P$ to two. Also we take into account random entries $(q,r) \notin S$. We ran experiments for $A$ with a matrix density of $|S| = 0.01n^2$ for $n = 1000, 5000$ and $10000$ and $m = 20$ with increasing $\alpha$ from 1 to 20 iteratively and measure a Hamming distance of at most $0.28\%$ of the problem size. To get a valid permutation matrix, we used the same heuristic as in the dense case and measured a relative error below $7.8\%$, compared to the Hungarian algorithm. Also, we could measure a memory reduction by over $65\%$.

### 4.5 Quadratic Assignment Problems

Quadratic Assignment Problems (QAPs) is a broadly employed mathematical tool for many real-life problems in operations research such as circuit design and shape matching. We demonstrate the application of our approach to non-convex QAPs of the form

$$\underset{V,W}{\arg\min}\, p(V,W)^T A\, p(V,W) \tag{14}$$

where $p(V,W) = \text{vec}(s\left(2\alpha VW^T\right))$ is the vectorized version of the permutation and $A \in \mathbb{R}^{n^2 \times n^2}$ is a cost matrix. $V$ and $W$ are normalized. The permutation matrix was optimized in a convex-concave manner, by optimizing the objective function

$$\underset{V,W}{\arg\min}\, p(V,W)^T (A - \beta I)\, p(V,W) + \mu(P(V,W)) \tag{15}$$

with $\beta$ being iteratively increased from $-\|A\|_2$ to $\|A\|_2$ and with $\mu(P(V,W))$ being the same permutation constraint regularizer as in (13).

We show results on the QAPLIB [6] library of quadratic assignment problems of real-world applications which range between $n = 12$ and $n = 256$ and we choose $m = \text{ceil}(\frac{n}{3})$. The problems in QAPLIB are meant to be challenging and optimal solutions for some of the larger problems are not provided because they are unknown. Thus, we report the gap to optimality (when known) of our solution and consider a solution good if it falls within $10\%$ of the optimum. We report the relative error and runtime in Fig. 3. In 75 out of 87 instances the result was a valid permutation matrix.

### 4.6 Shape Matching

Finally, we further assess the effectiveness of our approach for the the application of non-rigid shape matching, a common task in computer graphics and computer vision. To this end, we incorporate

Table 2: Geodesic errors and standard deviation (*std*) for noise-free and noisy data by Marin *et al.* [26] and our approach.

|       | $e_{prob}$ | $std\,(e_{prob})$     | $e_{emb}$ | $std\,(e_{emb})$      |            |
|-------|------------|-----------------------|-----------|-----------------------|------------|
| [26]  | 0.051      | $17.4 \times 10^{-4}$ | 0.029     | $3.5 \times 10^{-4}$  | noisy      |
| ours  | 0.047      | $26.9 \times 10^{-4}$ | 0.026     | $29.2 \times 10^{-4}$ |            |
| [26]  | 0.043      | $16.3 \times 10^{-4}$ | 0.022     | $3.5 \times 10^{-4}$  | noise-free |
| ours  | 0.041      | $8.1 \times 10^{-4}$  | 0.019     | $3.7 \times 10^{-4}$  |            |

our permutation matrix representation approach into the state-of-the-art shape-matching approach by Marin *et al.* [26], which learns the point correspondences using two consecutive networks $N_\theta$ and $G_\theta$, predicting shape embeddings and probe functions, respectively. We propose to replace the calculation of the permutation matrix based on the output of the first network $N_\theta$ by $s\left(\alpha V W^T\right)$, with $\alpha = 40$. The network transforms the vertices of 3D objects $X_x$ and $X_y$ into embeddings $\phi_x = N_\theta(X_x)$ and $\phi_y = N_\theta(X_y)$, which are used to compute $V = \phi_x(\phi_x^\dagger P_{gt}\phi_y)$ and $W = \phi_y$. $V$ here replaces a transformed embedding. The network is trained on the modified loss function

$$\min_\theta \sum_l \|s\left(2\alpha V W^T\right)^l X_y^l - P_{gt}^l X_y^l\|_2^2 \tag{16}$$

for a given ground truth permutation $P_{gt}$, and $V$ and $W$ being normalized row-wise. Similar to Marin *et al.* , we train the networks over 1600 epochs on 10000 shapes of the SURREAL dataset [45] and evaluate our experiments on 100 noisy and noise-free objects of different shapes and poses of the FAUST dataset [3], that are provided by [26] in [25].

We follow the evaluation of Marin *et al.* [26] and calculate the geodesic distance between the ground truth matching and the predicted matching $match_1 = \mathcal{N}(\phi_x C_1^T, \phi_y)$ for $C_1 = ((\phi_y^\dagger G_\theta(X_y))^T)^\dagger(\phi_x^\dagger G_\theta(X_x))^T$ whereby $\mathcal{N}$ is the nearest neighbor. In the following, we refer to the measured geodesic distance as $e_{prob}$. A second error ($e_{emb}$) which only concerns the first network's predictions, is measured by the geodesic distance towards $match_2 = \mathcal{N}(\phi_x, \phi_y C)$ for $C = \phi_y^\dagger P_{gt}\phi_x$, which is, again, calculated following Marin *et al.* [26]. The results of our experiments are reported in Table 2, showing the average geodesic errors (over 10 runs for each experiment) for the approach presented in [26] and our method. The table reveals improved results compared to [26].

**Stochastic Training.** Given that the explicit calculation of the permutation matrix in (16) is memory-intensive for a large number of vertices, we employ stochastic training to avoid the need for computing the full permutation matrix. As we describe in Sec. 4.1 we only calculate the loss over a few entries where the final permutation ought to be equal to one and on $k$ (here $k$ can be $\geq 1$) randomly chosen entries of each row of $P$ in each iteration. This approach reduces the memory requirement and gives us the possibility to train with larger shapes consisting of more vertices. In our experiments, we applied the stochastic training technique on the SURREAL dataset, and then evaluated the performance on FAUST by measuring the error rates for varying values of $k$, as depicted in Fig. 4b. We observed a small relative increase of less than $17\%$ in $e_{emb}$, and also a small effect on $e_{prob}$, but with a less clear tendency as one could see for $e_{emb}$. For $e_{prob}$ we measured an average standard deviation of $2.25 \times 10^{-3}$ and for $e_{emb}$ of $3.3 \times 10^{-4}$. Two noise-free examples of correspondences, visualized for full training and for stochastic training with $k = 1$, are shown in Fig. 4a, with the reference image on the left and the corresponding shapes on the right.

To evaluate the impact on the error and memory consumption when dealing with high-resolution objects (with more vertices), we ran experiments using data from the TOSCA dataset [5]. We train 400 epochs on the objects of the classes *victoria* and *michael* (32 objects in total) with up to 20000 sampled vertices. These experiments reveal a significant reduction of memory consumption, see Fig. 5a. Additionally, we evaluate the training on the class *david* and report a relative memory-error trade-off for up to $20k$ samples of each object, see Fig. 5b. The graph indicates a correlation between higher memory usage and lower error values for $e_{emb}$. The trends observed in the memory-error trade-off for $e_{emb}$ are generally applicable to $e_{prob}$ as well, although with some outliers.

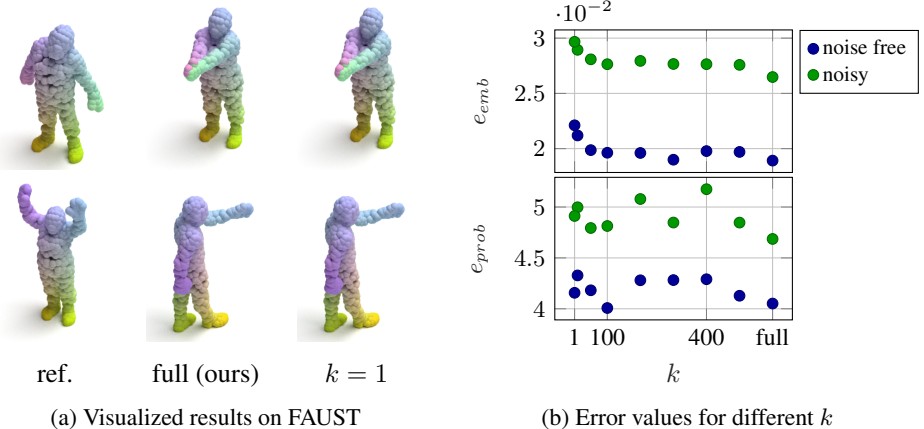

| | | |
|:---:|:---:|:---:|
| ref. | full (ours) | $k = 1$ |

(a) Visualized results on FAUST

(b) Error values for different $k$

Figure 4: Visualized matching results (a) and error values (b) for the FAUST dataset for different levels of sparseness $k$ during stochastic training

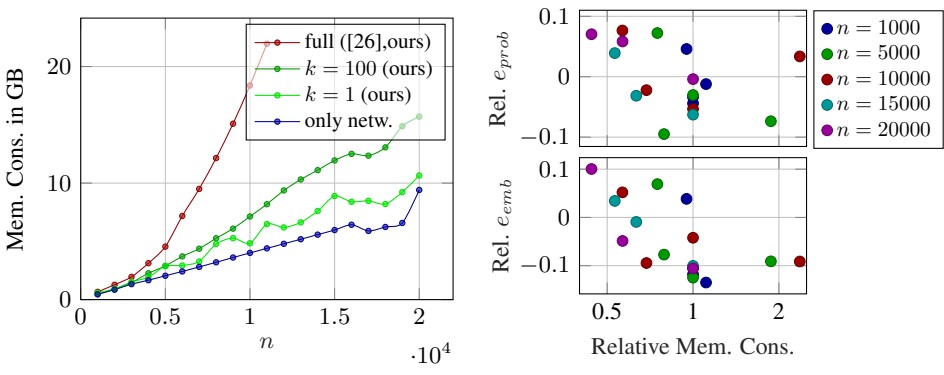

(a) Memory usage during non-stochastic training (red), stochastic training for different $k$ (green), and just calculating the network output (blue)

(b) Relative memory - error trade-off for a varying number of vertices $(n)$

Figure 5: (a) shows the memory consumption during training for shape matching for objects with a varying number of vertices $(n)$ and varying sparseness $(k)$. (b) shows the relative memory-error trade-off for varying sparseness (which causes the memory reduction), whereby the memory consumption is relative to $k = 100$ and the errors are relative to full training by [26] for $n = 1000$.

## 5 Conclusion

In this work, we proposed a strategy to represent permutation matrices by a low-rank matrix factorization followed by a nonlinearity and showed that by using the Kissing number theory, we can determine the minimum rank necessary for representing a permutation matrix of a given size, allowing for a more memory-efficient representation. We validated this method with experiments on LAPs and QAPs as well as a real-world shape matching application and showed improvements in the latter. Additionally, we explored the potential of optimizing permutations stochastically to decrease memory usage, which opens the possibility of handling high-resolution data.

**Limitations and Broader Impact**   Our method offers a promising solution to contribute positively to the environment by reducing the computational cost of a variety of problems involving permutation matrices. We do not see any ethical concerns associated with our approach itself. However, it is important to acknowledge a limitation of our method. For certain problem formulations, such as the Koopmans and Beckmann form QAPs, stochastic learning may not be feasible because the double occurrences of the permutation matrix make the stochastic computation not applicable. Moreover, our method requires devising a non-trivial, problem-specific adaptation.

## Acknowledgment

Yuval Bahat received funding from the European Union's Horizon 2020 research and innovation programme under the Marie Skłodowska-Curie grant agreement No 945422. Zorah Lähner is funded by a KI-Starter grant of the Ministry of Culture and Science of the State of North Rhine-Westphalia. Hannah Dröge acknowledges the support of the German Research Foundation (DFG) grant MO 2962/5-1.

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
