# Kissing to Find a Match:
# Efficient Low-Rank Permutation Representation - Supplementary Material

**Hannah Dröge**
University of Siegen
57076 Siegen, Germany
hannah.droege@uni-siegen.de

**Zorah Lähner**
University of Siegen
57076 Siegen, Germany
zorah.laehner@uni-siegen.de

**Yuval Bahat**
Princeton University
Princeton, NJ 08544, United States
yuval.bahat@gmail.com

**Onofre Martorell**
University of the Balearic Islands
Investigador ForInDoc del Govern de les Illes Balears
07122 Palma, Illes Balears, Spain
o.martorell@uib.cat

**Felix Heide**
Princeton University
Princeton, NJ 08544, United States
fheide@cs.princeton.edu

**Michael Möller**
University of Siegen
57076 Siegen, Germany
michael.moeller@uni-siegen.de

Our supplementary material includes a figure demonstrating the ability of our method to handle large matching problems and a graph showing the influence of the permutation matrix sparsity on the computation speed, as well as accuracy values on the experiments on point cloud assignment. As our method requires devising problem-specific adaptations, the supplementary material also includes a discussion on potential adaptations to our method. Further, it gives a short note on the non-linearity (ReLU) in our approach.

## Handling Large Permutation Matrices

Following our shape-matching experiments described in Sec. 4.6, we further visualize in Fig. 6 how the proposed approach enables handling large problems that would have been infeasible otherwise. The dashed red curve added to this figure (on top of the curves presented in Fig. 5a of the paper) corresponds to the estimated memory the would have been required to accommodate full permutation matrices, as a function of problem size $n$. While our approach can accurately handle large problems with as much as $n = 20,000$ vertices (green curves), running the equivalent experiments without it (red curves) would require prohibitively large amounts of memory ($\sim$ 73.6 GB, vs. 10.7 GB using $k = 1$). For estimating memory values (dashed curve) we assume memory usage follows a $c \cdot n^2$ curve, and estimate the value for $c$ based on the full matrix experiments (solid red) we conducted for $n \leq 11,000$.

## Influence of $k$ on Computation Speed

Further details regarding the influence of the variable $k$, that determines the sparsity of the calculated permutation matrix, on the computation speed is shown in Fig. 7. It shows the average computational speed per epoch (employing a batch size of 8) for the network $N_\theta$, used for the experiments discussed in Sec. 4.6. The recorded time values align with the accuracy measurements presented in Fig. 4b.

37th Conference on Neural Information Processing Systems (NeurIPS 2023).

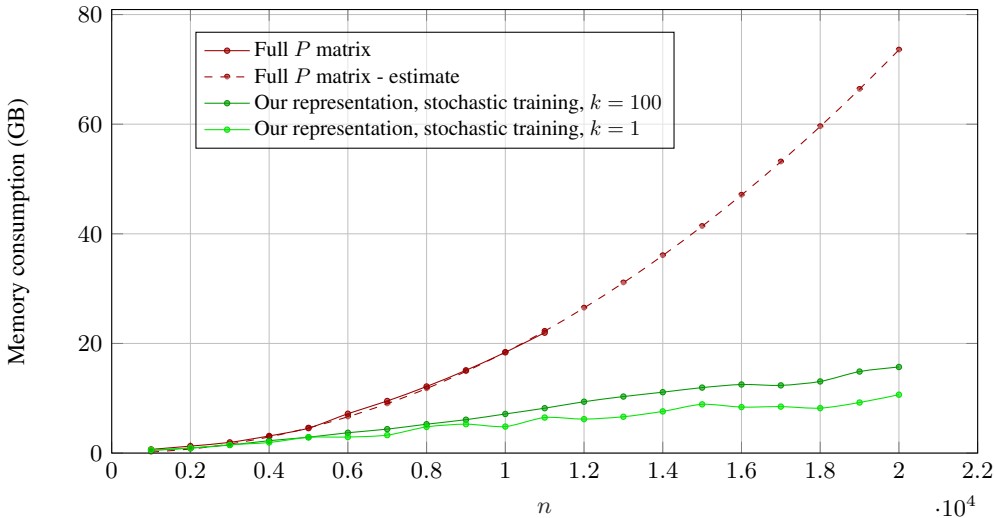

Figure 6: **Memory savings.** Memory usage when training the shape matching network of [26] with different permutation matrix representations: Using full matrices (red) vs. using our stochastic training scheme with different sparseness levels (green). See text for details.

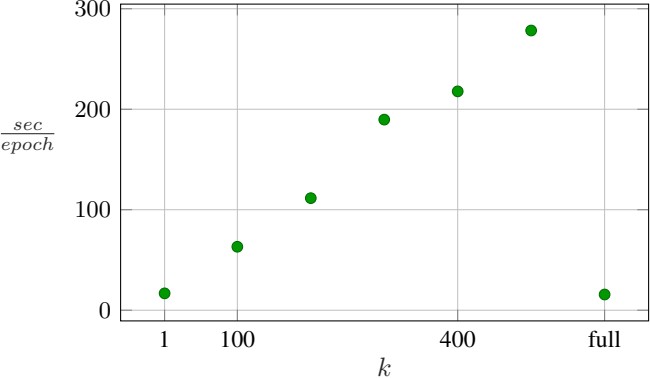

Figure 7: **Time measurement.** Average time (in seconds) needed to optimize the shape matching network $N_\theta$ per epoch, depending on the stochastic variable $k$.

## Accuracies for Point Cloud Assignment Experiments

Following our experiments on Point Cloud Assignment in Sec. 4.2, we add additional accuracy values to the prediction of a linear transformation over point clouds in Table 1. Here, we measure the distance between the true point clouds and their transformed counterparts, for each problem size ($n$).

| nonlin.\$n$ | 10 | 100 | 1000 | 10000 |
|---|---|---|---|---|
| ReLU | $2.362 \times 10^{-4}$ | $1.0597 \times 10^{-4}$ | $4.115 \times 10^{-4}$ | $1.0387 \times 10^{-4}$ |
| SoftMax | 0.0829 | 0.026 | 0.0057 | 0.0012 |
| Stoch. Softmax | 0.0712 | 0.0164 | 0.00173 | 0.0002 |

Table 1: $\ell_2$ norm distance between true and transformed point clouds in the point cloud alignment experiment across various non-linearities, and problem size ($n$).

## Individual Adaptions

Adaptions to our method can involve the learning rate, as well as the selection of the $\alpha$-parameter in the equation of Proposition 2. If talking about Equation 6, one can consider the adaptation of the thresholding (for the equation $2\sigma(2VW^T - 1)$ the threshold is set to 1). By decreasing the threshold, we simplify the optimization process, as fewer gradients are excluded from the experiment, while this could result in a less precise outcome. Additional adaptions might concern optimization techniques, such as fixing one matrix with descent characteristics (e.g. Gaussian random) in order to simplify the optimization. Moreover, it's possibly also necessary to adapt a network architecture that predicts the matrices $V$ and $W$, and with further research in this direction, we believe to expand the potential to provide memory reduction benefits.

## Notes on the Nonlinearity

The ReLU merely serves as a type of thresholding operation and could be replaced by any other function that is zero for all values below a certain threshold and one for an input value of one. In fact, an arbitrarily low rank can still allow to represent any permutation exactly by letting the threshold approach one. Yet, since gradients of any entry below the threshold are zero, such a representation becomes increasingly difficult to optimize.