# OpenReview forum: "Kissing to Find a Match: Efficient Low-Rank Permutation Representation"
_NeurIPS.cc/2023/Conference — NeurIPS 2023 poster_

### Official Review · Reviewer_tWZ6 · 2023-06-29

**Soundness:** 4 excellent
**Presentation:** 4 excellent
**Contribution:** 4 excellent
**Rating:** 8
**Confidence:** 4

**Summary:**

The authors propose a provably exact and an approximative method for calculating permutation matrices requiring much lower memory than previous approaches. The algorithms are based on Kissing numbers and describe row relationships as their cosine value. This allows for representing the problem with significantly fewer values than the baseline n*n permutation matrix representation. Additionally, the authors propose a relaxation of this exact algorithm, utilizing the SoftMax operator, that alleviates the issues with the optimization for large problems. The authors also show a number of applications where their method is applicable.

**Strengths:**

I believe this is a pretty strong work with a very general and useful contribution. Strengths in more details:
- A provably exact algorithm that significantly reduces the memory consumption of general permutation matrices.
- Additionally, a relaxation is proposed that has better convergence properties in optimization.
- The authors demonstrate on a number of applications that their method is applicable and useful.

**Weaknesses:**

First, I wanted to give a strong accept, but the experiments left me slightly in doubt.
My main problem is the following: The experiments focus a little bit too much on the application side (which is amazing), but miss to do a general comparison to baselines and the proposed methods. I would have liked to see comparisons to SOTA algorithms (related works mention many) on finding permutation matrices, showing run-time, memory, and errors. While the exact algorithm will probably have 0 error, some other methods might have a little bit bigger while being orders of magnitude faster. Also, to my understanding, the exact algorithm breaks down with the problem size, when the SoftMax version is used, which is not exact anymore. Also, it would be good to see the breaking point of the exact algorithm where it does not converge anymore. In brief, a thorough comparison to the SOTA is missing.

I still like the paper, but without such comparison (considering the theoretical value), I do not give a strong accept. If the authors can provide such an analysis to understand the trade-offs in their rebuttal, I consider improving the rating.


**Questions:**

Experiments:
- L191 is translation not considered on purpose?
- L204-209 This is a little but unclear. Does the propose method always, without failure, find the correct permutation and transformation? If this is the case, this should be highlighted more. If not, an error value should be shown.
- L209 Equally good results with Softmax. Does this mean that the approximation in this noise-free case can be compensated?
- L215 refers to Fig.2, but that seems to show something different (Dense/Sparse) and not the topic the authors mention at this line.

Typos:
L135 "either exact" -> "either as exact"
L156 "valdidate" -> "validate"
L179 missing dot from the end of the sentence
L230 "similariy" -> "similarity"


**Limitations:**

Discussed.

---

> ### Author Rebuttal · Authors · 2023-08-09
>
> Regarding the request for comparison to baseline methods, please refer to the general response to all reviewers, where we have shown comparisons to baseline methods within the functional maps framework [30].
>
> #### Question 1
> L191 is translation not considered on purpose?
> #### Answer 1
> Yes, we didn’t consider translations, as the experiments were primarily focused on a proof of concept, and translation can be removed through centering.
>
>
> #### Question 2
> L204-209 This is a little but unclear. Does the propose method always, without failure, find the correct permutation and transformation? If this is the case, this should be highlighted more. If not, an error value should be shown.
> #### Answer 2
> In these experiments the proposed method always finds the correct permutation, and with it also the correct assignment of the transformed vectors. We will further highlight this in the final version. Nevertheless, we have small shiftings in the transformation of the vectors, for which we will add the accuracies (see the general response to all reviewers).
>
> #### Question 3
> L209 Equally good results with Softmax. Does this mean that the approximation in this noise-free case can be compensated?
> #### Answer 3
> Yes, for finding the correct assignment in this example, this is the case.
>
> #### Question 4
> L215 refers to Fig. 2, but that seems to show something different (Dense/Sparse) and not the topic the authors mention at this line.
> #### Answer 4
> In this line, we referred to the memory reduction, which is shown in Fig 2. But we agree with the reviewer that this point does not become clear in the submitted version. We will rewrite this part of the text to ensure that it is correctly understood.

---

> > ### Comment · Reviewer_tWZ6 · 2023-08-17
> >
> > I appreciate the authors' answers and the baseline comparison. I like the paper and improve my rating to strong accept.

---

### Official Review · Reviewer_2QBr · 2023-07-04

**Soundness:** 2 fair
**Presentation:** 4 excellent
**Contribution:** 2 fair
**Rating:** 4
**Confidence:** 4

**Summary:**

This paper proposes a formulation for representing high-dimensional permutation matrices. The basic idea is to express the n x n permutation matrix as an elementwise non-linear function of a product of low-rank matrices V,W (n x m). The authors introduce the kissing number theory which gives the minimum number 'm' for which such a decomposition is provably possible and then show intuitive proofs that in scenarios when the decomposition is valid - their construction is appropriate. In addition, they also promote two well-known non-linearities (ReLU and SoftMax) which can be used in the optimization process for the representation.

Results are demonstrated on two synthetic and one recent 3D shape-matching scenario. Broadly the experiments make an impression of a significantly reduced memory consumption and the ability to recover permutations. In the shape-matching example, the proposed formulation also slightly improves on the previous baseline.

**Strengths:**

- The core idea of this paper (low-rank representation of permutations with nonlinearities) is *very* interesting and potentially has a very wide impact in scenarios where matching between pointsets is a crucial problem (linear assignment, quadratic assignment etc)
- Overall the writing of this paper is very good, and the background and related material on the kissing number theory has been introduced and explained in a way interesting way.
- The benefits of the proposed construction are (1.) strong memory reduction in representing permutation matrices (2.) An accompanying optimization scheme that allows for recovering permutations

**Weaknesses:**

- Broadly, I felt the experimental section is very rudimentary. There are very few comparisons to conceptual baselines: namely using stochastic matrices, optimal transport, Hungarian algorithm and nearest neighbor-like methods. It is not clear from the experiments whether the obtained solutions are: not just some permutations, but the *correct* permutations.

- More specifically, for the point cloud example in section 4.2, what is the accuracy of the recovered transformation $\Theta$ as a function of n? How do other conceptual baselines compare in this example both in terms of accuracy and memory complexity?

- Despite the experiments on Marin et.al, perhaps a simpler and more convincing demonstration would be to use the proposed permutation representation in either or all of [43], [30], etc. where a linear/quadratic assignment is solved and then compare with the original methods (perhaps yielding an improvement in memory with a comparable or better accuracy)


**Questions:**

See Weaknesses

**Limitations:**

I do not think this paper has any direct negative societal impact. Please see the weaknesses for technical limitations.

Overall, this paper has a very interesting idea and a clever conceptual message on representing permutations. My biggest concern is whether the proposed construction is impactful in terms of ease of optimization and acceptable accuracies for the multitude of shape-matching problems that this can be applied to. Given the lack of comparisons to conceptual baselines (i.e. not in terms of a state-of-the art shape matching paper, but even simply comparing with previous permutation representations like stochastic matrices, or spectral decompositions - in any framework), I am inclined towards a weak reject at this point. I can be convinced of the gains in memory complexity of the proposed representation but am not yet sold on its applicability and accuracy.

---

> ### Author Rebuttal · Authors · 2023-08-09
>
> Regarding the request to include conceptual baselines to compare in the point cloud example, we have included experiments that can be found in the general response to all reviewers under the topic "Comparisons with Permutation Baselines in [30]". These experiments include comparisons in accuracy, time, and memory performance for extracting point-wise correspondences, compared to the LAP solver, nearest-neighbors, optimal transport, and sinkhorn iteration, within the functional maps framework [30].
>
>
> #### Question 1
> It is not clear from the experiments whether the obtained solutions are: not just some permutations, but the correct permutations.
> #### Answer 1
> In the LAP and QAP experiments, we evaluated the permutation matrix from of the relative error of the energy compared to the energy of the optimal solution. We will make this clear in the final version.
>
> #### Question 2
> More specifically, for the point cloud example in section 4.2, what is the accuracy of the recovered transformation as a function of $n$?
> #### Answer 2
> We add the accuracy values of the transformation (depending on $n$) in the form of measuring the distance between the true and the transformed point clouds to the final version (see the general response to all reviewers for the accuracy values).

---

### Official Review · Reviewer_KuKh · 2023-07-04

**Soundness:** 4 excellent
**Presentation:** 4 excellent
**Contribution:** 4 excellent
**Rating:** 9
**Confidence:** 4

**Summary:**

The paper proposes a novel approach for representing permutation matrices with low-rank matrix factorisation. The method employs Kissing number theory to find the minimum rank necessary to represent the target matrix. This is often quite a bit lower than the rank of the original matrix, so allows for a more efficient memory representation. The paper also shows how the approach can be used in practice in various relevant problems, such as point cloud alignment or shape matching.

Edit: I have read the rebuttal, and given the scores from other reviewers and I would like to keep my score.

**Strengths:**

* The work deals with a meaningful problem, and produces an elegant solution with wide ranging applicability.
* The authors demonstrate the performance of their approach in various problems, demonstrating similar / better accuracy with a significant improvement in memory requirement.
* The paper is very well written and easy to read.
* The experiments are relevant.
* The formulation is principled and includes several relevant proofs.

Overall I believe this is a perfect NeurIPS paper. While I have little knowledge of Kissing number theory, the paper solves a very relevant problem in an elegant way, and demonstrates impressive and wide-ranging practical applicability in multiple domains. I therefore strongly believe the work should be accepted.

**Weaknesses:**

I have very little negative notes about the work, though, I could say that the ablation section could be expanded, e.g. by seeing how increasing the stochastic training k affects speed and accuracy.

**Questions:**

none

**Limitations:**

These are addressed at the end of the paper.

---

> ### Author Rebuttal · Authors · 2023-08-09
>
> We would like to thank you for your very positive response to our work. We have added an analysis of the relationship between $k$ and the optimization time to the general response.

---

### Official Review · Reviewer_1Eor · 2023-07-06

**Soundness:** 4 excellent
**Presentation:** 4 excellent
**Contribution:** 4 excellent
**Rating:** 8
**Confidence:** 2

**Summary:**

This work addresses the problem of estimating large permutation matrices by approximating them with a low-rank factorization follow by a nonlinear mapping, thereby reducing the storage complexity from O(n^2) to O(n).

The main contribution of the paper is i) a theoretical derivation and proof of the minimal required rank of the factorization matrices, and ii) the use of a nonlinearity, such that the full rank permutation matrix can be restored *exactly*. i) is based on the Kissing number (or bounds on it), while the nonlinearity in ii) can be a ReLu. Importantly, the possibility to exactly represent any permutation matrix is in strong contrast to direct low-rank factorizations which can only approximate the permutation matrix (since unable to recover the full rank).

The authors propose practical solutions for the optimization: A smooth version of the nonlinearity (softmax), and an optimization scheme (inspired by stochastic optimization; gradients are considering a row of the factorization matrices at a time) that approximates the full objective, but never requires to build the full permutation matrix, thus fully leveraging the compact representation and resulting storage savings.

Experimental results are extensive and demonstrate the applicability of the method on point cloud alignment, linear and quadratic assignment problems, and shape matching.

**Strengths:**

**S1** Proposition 1 and Eq. (7) represent a novel contribution. The insight that a non-linear mapping of a matrix factorization is able to recover the full rank permutation matrix is a strong contribution and is of interest for the wider community.

**S2** The proof of the minimal require factorization matrix rank via the Kissing number is an interesting theoretical contribution. Also practically it provides a clear guidance on the required size of factorization matrices such that the exact permutation matrix can be recovered.

**S3** The proposed stochastic optimization in Sec. 4.1 provides a practical algorithm for optimizing the permutation matrix, while leveraging the compact representation (the full permutation matrix is never built, but only computed element-wise at a time). The non-smoothness of the ReLU is addressed by soft-max function with a controllable temperature parameter that allows to approximate the exact solution with desired accuracy over the course of the optimization procedure.

**S4** Experimental results are performed across different application domains and demonstrate the usefulness of the proposed approach.

**Weaknesses:**

None, but I'm also not an expert in the area. I'm especially not knowledgeable about related work and thus can not judge the novelty of the proposed approach.

**Questions:**

**Q1** Is there a relation between the Kissing number and the ReLU? Is it conceivable that there exists another non-linear function that allows an even lower rank factorization while still being able to represent that exact permutation matrix?

**Limitations:**

-

---

> ### Author Rebuttal · Authors · 2023-08-09
>
> #### Question
> Is there a relation between the Kissing number and the ReLU? Is it conceivable that there exists another non-linear function that allows an even lower rank factorization while still being able to represent that exact permutation matrix?
>
> #### Answer
> The ReLU merely serves as a type of thresholding operation and could be replaced by any other function that is zero for all values below a certain threshold and one for an input value of one. In fact, an arbitrarily low rank $\geq 2$ can still allow to represent any permutation exactly by letting the threshold approach one. Yet, since gradients of any entry below the threshold are zero, such a representation becomes increasingly difficult to optimize (please also see the answer for reviewer AkAs). We will add a discussion on this aspect in the revised version of this paper.

---

### Official Review · Reviewer_AkAs · 2023-07-06

**Soundness:** 3 good
**Presentation:** 3 good
**Contribution:** 2 fair
**Rating:** 5
**Confidence:** 3

**Summary:**

This paper proposes a novel way to decompose a permutation matrix into two low-rank matrices so that a significant amount of  space can be saved to store a permutation. Then the authors further implement such decomposition to solve practical tasks of point cloud alignment, assignment problem and shape matching.

**Strengths:**

Overall, the paper is well-written and clearly explains the heuristics behind the method. The method of using Kissing numbers and non-linearity to perform low-rank decomposition of a permutation matrix is very interesting and can possibly inspire future research. Based on Fig 5, the method indeed provides memory saving for supervised learning.

**Weaknesses:**

The most troubling weakness of this method lies in its value of application. Specifically, I have the following concerns:

1. The method introduces two complex problems to optimization: bi-variable structure and non-linearity. As mentioned by the authors, this requires devising a non-trivial, problem-specific adaptation for each problem.

2. To avoid forming an $n\times n$ matrix, the authors propose to only optimize over a handful of entries, which basically requires knowing the ground truth permutation (or assuming some sparse structures as in LAP). This leads to a very limited set of application scenarios.


**Questions:**

1. Kissing number is based on a threshold of arccos(0.5), does this mean that if we switch to a smaller angle, we can fit more vectors in a unit sphere, thus save more space to store a permutation? If so, why don't we do so?

2. When both $V$ and $W$ need to be optimized, are they optimized in parallel or alternatingly? Specifically which optimization algorithm is used for your numerical tests?

3. How do you implement Stochastic Optimization for LAP and QAP when $A$ is dense?

4. In the part (b) of Fig 5, what is the scale of y axis? The term "Relative" here is very confusing and without explanation.


**Limitations:**

The limitations of the work are mostly mentioned in the weakness section. The authors indeed addressed them but it is still quite confusing to read for the first time.

---

> ### Author Rebuttal · Authors · 2023-08-09
>
> #### Question 1
> Kissing number is based on a threshold of arccos(0.5), does this mean that if we switch to a smaller angle, we can fit more vectors in a unit sphere, thus save more space to store a permutation? If so, why don't we do so?
> #### Answer 1
> Yes, it is correct that the threshold does not need to be 0.5. In fact, if we pick any $V \in \mathbb{R}^{n \times m}$, $m<<n$, with normalized rows that are non-repeating, then the threshold $\mu := \max_{i\neq j} \langle V_{i,:} , V_{j,:} \rangle$ allows to use our approach for representing a permutation $P$ by choosing $W=PV$. Even for Gaussian random matrices, this allows to reduce the rank exponentially, see e.g. "Limiting Laws of Coherence of Random Matrices with Applications to Testing Covariance Structure and Construction of Compressed Sensing Matrices." by Cai and Jiang, where it is shown that $\mu$ behaves like $2\sqrt{\frac{\log(n)}{m}}$ if $n$ is an exponential of $m$. Yet, the ReLU indeed acts as a thresholding such that thresholds approaching 1 make it extremely difficult to still optimize the resulting objective. We found a threshold of $0.5$ to yield a good compromise between the ability to optimize (with first-order methods) and an accurate representation of permutations via a low-rank factorization while benefitting from some additional literature deriving theoretical bounds on the factorized dimension. Adaptive thresholds (e.g. using continuation schemes) and/or softmax approaches with iteratively increasing temperature (converging from soft- to hard-(arg)-max) are of course possible fine-tuning options to improve results in particular applications.
>
>
> #### Question 2
> When both V and W need to be optimized, are they optimized in parallel or alternatingly? Specifically which optimization algorithm is used for your numerical tests?
> #### Answer 2
> The estimation of V and W is performed in parallel using the Adam optimization algorithm. We will clarify this aspect in the final version of the paper. We will also upload our code, and make it publicity available upon acceptance.
>
> #### Question 3
> How do you implement Stochastic Optimization for LAP and QAP when $A$ is dense?
> #### Answer 3
> For LAP the similarity matrix $A$ has the dimension $n \times n$ (the same dimension as the permutation matrix), for QAP the similarity matrix is even larger with $A \in \mathbb{R}^{n^2 \times n^2}$, or for the Koopmans and Beckmann formulation we have two matrices of the dimension  $n \times n$. If those matrices have to be computed densely, it would require as much memory as a fully calculated permutation matrix or even more, so the stochastic optimization would not make any difference. Yet, considering that the costs are still sums over many terms, stochastic/alternating optimization schemes that only consider a few terms at a time and compute costs on-the-fly could be implemented. We have not tested our approach in this respect yet.
>
> #### Question 4
> In the part (b) of Fig 5, what is the scale of y axis? The term "Relative" here is very confusing and without explanation.
> #### Answer 4
> The term “relative” means, that the errors are calculated relative to the error of the full training by [26] for $n = 1000$, therefore it is calculated by $error_{relative} = \frac{error_{ours} - error_{[26]}}{error_{[26]}}$.
> We will make this more clear in the final version.
>
> #### Unknown Ground Truth and Limitations
>
> - To avoid forming an matrix, the authors propose to only optimize over a handful of entries, which basically requires knowing the ground truth permutation (or assuming some sparse structures as in LAP). This leads to a very limited set of application scenarios.
> - The limitations of the work are mostly mentioned in the weakness section. The authors indeed addressed them but it is still quite confusing to read for the first time.
>
>
> #### Answer
> We will make the limitations of our approach more visible by collecting them in a separate section in the final version of our paper. The approach of (sparse) memory-saving optimization is applicable to any fully supervised approach for learning permutation matrices. Additionally, we anticipate that extensions to self-supervised settings like linear assignment problems are possible by turning to stochastic/alternating optimization schemes that do not consider all terms of the cost function at a time.

---

> > ### Comment · Reviewer_AkAs · 2023-08-15
> >
> > I thank the authors for answering my questions. However, the weakness I mentioned in my review remains. Specifically, the method can only be implemented in supervised learning scenarios, missing out on a wide range of applications. As a result, I will not change my current rating.

---

> > > ### Author Response · Authors · 2023-08-15
> > >
> > > Thank you for your consideration! While we are hopeful that an extension to a fully unsupervised setting is possible, we consider supervised machine learning to be an extremely important topic for the NeurIPS conference audience and would like to point out, that a large number of learning techniques published at NeurIPS are fully supervised techniques (only).

---

### Author Rebuttal · Authors · 2023-08-09

We thank all reviewers for their comments that helped us improve the presentation of our work. In the following, we address the concerns that have been raised, and we provide additional experimental results:


# Comparisons with Permutation Baselines in [30]
We conduct an additional experiment where we compare different classical optimization approaches and to point cloud alignment experiment (see Section 4.2) in the context of the functional maps framework [30].
We compare our method, that calculates correspondences the same way as described in the Point Cloud Alignment experiment in Section 4.2., to
a general linear assignment problem (LAP) solver (specifically the Jonker-Volgenant algorithm from sklearn.linear\_sum\_assignment, which is in practice faster than the Hungarian algorithm), nearest neighbor computation, optimal transport (as implemented in the python POT package), and stochastic matrices generated by Sinkhorn iterations.

The experiment setup is as follows (all details will be included in the final version):
The goal is to extract a point-to-point correspondence between two shapes $X, Y$ from a $m \times m$-dimensional functional map [30] where $m$ is much smaller than the number of vertices in $X$ and $Y$.
A possible way to do this is by doing a nearest neighbor (NN) search of the spectral point representations $\Phi_X, \Phi_Y \in \mathbb{R}^{n \times m}$ aligned by the functional map as proposed in the original paper (see [30, Section 6.1]).
However, it is also possible to find an assignment between all rows of $\Phi_X, \Phi_Y$ by other means, for example by solving a linear assignment problem if a bijection is desired.
This is exactly the point cloud alignment setting from our experiments in Section 4.2 with a small amount of noise in the point clouds, and we show that our method outperforms all baselines in terms of geodesic error of the final matching and shows positive trends in terms of runtime and memory consumption.
We use the FAUST registrations [3] with the original $6890$ vertices and a downsampled version to $502$ vertices for those experiments.
The rows of $\Phi_X, \Phi_Y$ are generated by applying the ground-truth functional map to the spectral embedding of each vertex and can be assumed to be permuted, noisy versions of each other such that we can directly use them as $V$ and $W$ for our method.

We will add more details about the exact setup for all methods in the final version. The results of this comparison can be seen in Table 1 in the response PDF.
Our method has the best runtime/memory ratio on the higher dimensional example apart from nearest neighbors (which is known to become quite unreliable if the alignment is not tight) while still providing the most accurate results.







# Clarification of the Method Strengths
Our work proposes a novel efficient approach for representing permutations, which inherently provides a guaranteed minimal decomposition rank $m$ (and therefore the minimal memory requirements) necessary for representing a problem of size $n$. This allows tackling large problems whose size $n$ could not be handled by existing methods.
We believe that the strength of our method lies in its ability to efficiently represent permutation matrices while still being differentiable and enabling techniques such as sparse/stochastic optimization, which will be particularly efficient in a supervised learning setup.
However, it is not a plug-and-play solution that can be quickly incorporated into any pipeline.
Therefore, a direct but fair comparison to other learning-based permutation predictors like Sinkhorn layers is not straightforward to implement but requires individual adaptions of our method for each setting.
We showed that it is possible to considerably improve memory requirements by including it in Marin et al. [26], and we strongly believe this is possible in other methods with more research in this direction.


# Further Experimental Details on Point Cloud Alignment and Shape Matching
We include additional accuracy values on the prediction of a linear transformation over point clouds, by measuring the distance between the true point cloud and its transformed counterpart. We add these additional accuracy values for each problem size ($n$) which were previously outlined in the initial version of our work (see Table 2 in the response PDF).
To further expand the ablation study, we will include a comparison of the training speed for stochastic optimization in the shape-matching experiment, depending on the stochastic variable $k$, to the results in Figure 4b. The values of this additional experiment can be found in Table 3 in the response PDF.

---

> ### Comment · Reviewer_2QBr · 2023-08-14
> **Rebuttal Response**
>
> Thank you for the detailed rebuttal and the additional experiments, which is quite illuminating. However, I have a couple of concerns:
>
> (1.) In the additional experiment in Table 1, you assume the aligned spectral embeddings (i.e. after applying the linear transformation of the ground truth functional map)  to be directly used as V and W - do they satisfy the mandatory equalities and inequalities of Equation 6?
>
> (2.) For the results shown in Table 1 of the additional pdf document, if V and W are fixed, what do you optimize for in this case?
>
> (3.) I appreciate the clarity and honesty in declaring that the proposed method is *not* currently posed as a plug-and-play solution. Can you say something more about "individual adaptations in each setting?" I am thinking along - if I were to use your method, for some different setting how should I think of modifying equation 7?

---

> > ### Author Response · Authors · 2023-08-15
> >
> > In order to meet the constraint (of eq. 6) that the row-norm of both $V$ and $W$ equals one, we apply row-wise normalization (after transforming $\Phi_Y$). This ensures that the inequality holds once the correct alignment of the points is identified.
> >
> > In our method, optimizing $V$ and $W$ isn't always required. For instance, while in the LAP experiment, we optimize both $V$ and $W$, in this particular experiment, we focus on optimizing a transformation matrix. The transformation matrix is applied to the spectral point representations, $\Phi_Y$, on top of the functional map, aiming to derive $W$, similar to the point cloud alignment experiment.
> >
> > Adaptions to our method can involve the learning rate, as well as the selection of the $\alpha$-parameter in the equation of Proposition 2. If talking about Equation 7, one can consider the adaptation of the thresholding (for the equation $\sigma(2V W^T − 1)$ the threshold is set to 1). By decreasing the threshold, we simplify the optimization process, as fewer gradients are excluded from the experiment, while this could result in a less precise outcome.
> > Additional adaptions might concern optimization techniques, such as fixing one matrix with descent characteristics (e.g. Gaussian random) in order to simplify the optimization. Moreover, it's possibly also necessary to adapt a network architecture that predicts the matrices $V$ and $W$, and with further research in this direction, we believe to expand the potential to provide memory reduction benefits. We will add this information to our limitations.

---

> > > ### Comment · Area_Chair_9pKZ · 2023-08-18
> > >
> > > Dear Reviewer 2QBr,
> > >
> > > Thanks for actively engaging in a discussion with the authors. Did the authors' response address your comments?
> > >
> > > Best,
> > > AC

---

> > > > ### Comment · Reviewer_2QBr · 2023-08-18
> > > > **Re: Official Comment by Area Chair 9pKZ**
> > > >
> > > > I think there needs more work to apply the method impactfully. I appreciate the broad strategies outlined in the authors reply here, but it is not any concrete recipe and to be very honest this does bother me. However, I still stand by some very positive aspects of this paper: the idea is very novel and refreshing, the paper is written and messaged well and finally the experiments in the rebuttal have positive results very much in line with what I was hoping. I would give this paper a borderline accept and would not mind accepting it, but also keep enough space for calibrating this submission against an overall Neurips threshold since it is not a full hearted strong accept for me.

---

### Decision · Program_Chairs · 2023-09-21

**Decision:**

Accept (poster)

**Comment:**

This paper studies the low-rank representation of the permutation matrices, which have applications in robotics and computer vision, among others. One of the main challenges in the optimization problems defined over permutation matrices is the quadratic dependency of the decision variables on the problem dimension, which makes them prohibitive to solve in massive-scale instances. The authors provide an intuitive, yet elegant approach to alleviate this challenge. They show that a low-rank factorization followed by a ReLU activation layer can capture the set of all permutation matrices. Using this representation, they show that the optimization problems defined over permutation matrices with n^2 variables can be reduced to optimization problems defined over nm variables, where m can be significantly smaller than n. Their extensive numerical experiments showcase the empirical performance of their proposed idea in various settings.

Five reviewers have reviewed the paper, and their overall assessment of the paper was mostly positive. I agree with this assessment and believe the paper provides interesting and useful ideas to improve the complexity of solving optimization problems defined over permutation matrices.

The authors are highly encouraged to take into consideration the comments made by the reviewers. The AC also suggests that the authors include a brief discussion on the exactness of their relaxation, especially in settings where the proposed low-rank factorization +  ReLU introduces infeasible optimal solutions (i.e., optimal solutions that do not correspond to a valid permutation matrix).